# Measurement of the $t$-channel single top-quark production cross section at $\sqrt{s} = 13$ TeV with the ATLAS detector and interpretations of the measurement

**Maren Stratmann[1]⋆, on behalf of the ATLAS Collaboration**

**1** University of Wuppertal, Wuppertal, Germany

⋆ maren.stratmann@cern.ch

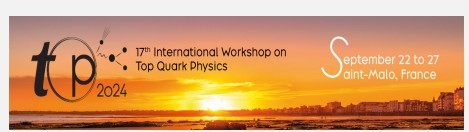

*The 17th International Workshop on
Top Quark Physics (TOP2024)
Saint-Malo, France, 22-27 September 2024*

## Abstract

The $t$-channel is the dominant production channel for single top-quarks at the LHC. The total cross section of this process is measured by ATLAS in proton-proton collisions at a center-of-mass energy $\sqrt{s} = 13$ TeV. The production cross sections for single top-quarks and single top-antiquarks are measured to be $\sigma_{tq} = 137^{+8}_{-8}$ pb and $\sigma_{\bar{t}q} = 84^{+6}_{-5}$ pb. For the combined cross section and the ratio of the cross sections $R_t$, $\sigma_{tq+\bar{t}q} = 221^{+13}_{-13}$ pb and $R_t = 1.636^{+0.036}_{-0.034}$ are obtained. As interpretations of the measurement, limits are set on the EFT operator $O^{3,1}_{Qq}$ and on the CKM matrix elements $|V_{td}|$, $|V_{ts}|$ and $|V_{tb}|$.

## 1 Introduction

The main production channel for single top-quarks at the LHC is the $t$-channel exchange of a virtual $W$ boson. With a precise measurement of the process, the Standard Model (SM) prediction can be tested and potential contributions from physics beyond the SM can be probed. The ratio $R_t = \sigma_{tq}/\sigma_{\bar{t}q}$ provides sensitivity to the predictions of different PDFs and the total cross-section can be used to set constrains on the Effective Field Theory (EFT) operator $O^{3,1}_{Qq}$. These proceedings summarize the results of the latest measurement of the $t$-channel production cross section by ATLAS [1] using the full Run 2 dataset collected during the years 2015-2018.
All graphics and numbers given in these proceedings are taken from Ref. [2].

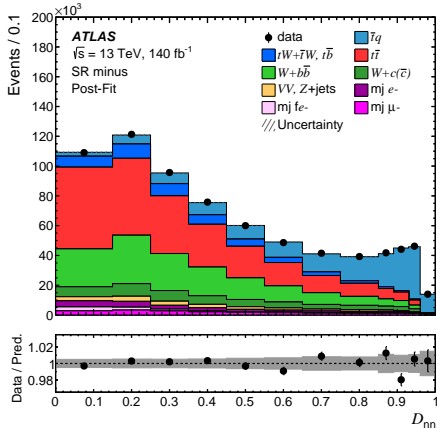

Figure 1: $D_{nn}$ distribution in the signal region for negative lepton charge after the maximum likelihood fit is performed. The gray band in the lower panel indicates the post-fit uncertainties [2].

## 2 Total cross section measurement

The events in the signal region are selected according to the expected signature of leading-order $t$-channel events. Exactly one charged light lepton (electron or muon), high missing transverse energy and exactly two jets are required. For the jets, exactly one of them has to be $b$-tagged using the DL1r [3] tagger with a 60% working point. Additional requirements are imposed to reduce the contribution of background events in the signal region. Two separate signal regions are defined according to the electrical charge of the lepton.

To seperate signal and background events, a feed-forward Neural Network (NN) is trained on 17 input variables. A good discriminating power between signal and background events is achieved, as it can be seen from Figure 1. Signal events are classified with high values of the NN output score $D_{nn}$, while background events are sorted towards low $D_{nn}$ values. The cross sections are determined by performing a binned maximum likelihood fit to the $D_{nn}$ distributions. The distribution in Figure 1 shows the post-fit result. A good agreement between the MC simulation and the data is reached.

The obtained results for the cross sections are $\sigma_{tq} = 137^{+8}_{-8}$ pb and $\sigma_{\bar{t}q} = 84^{+6}_{-5}$ pb. The combined cross section and $R_t$ are measured to be $\sigma_{tq+\bar{t}q} = 221^{+13}_{-13}$ pb and $R_t = 1.636^{+0.036}_{-0.034}$. A comparison of the result for $R_t$ with the predictions from different PDF sets is presented in Figure 2. Most predictions are compatible with the measurement within the uncertainties.

## 3 Interpretations of the measurement

As interpretations of the measurement, constrains are set on the EFT operator $O_{Qq}^{3,1}$ and on the CKM matrix elements $|V_{tx}|$.

### 3.1 EFT interpretation

The EFT operator $O_{Qq}^{3,1}$ introduces a four-fermion contact interaction involving two quarks of the third generation and two quark of the first or second generation. A non-zero contribution from this operator affects the top-quark production angle and therefore changes the fiducial acceptance $\mathcal{A}$ of the signal events. The signal events $\mathcal{A} \times \sigma$ depends quadratically on the Wilson coefficient $C_{Qq}^{3,1}/\Lambda^2$, as shown in [5]. This dependence is used for the interpretation.

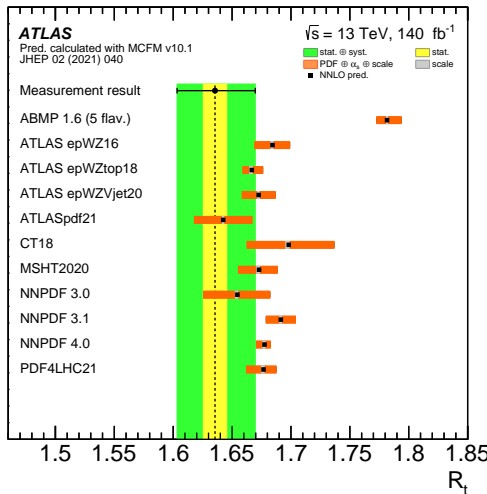

Figure 2: Measurement result for $R_t$ compared to the NNLO predictions calculated with MCFM [4] using different PDF sets [2].

Dedicated MC samples are produced with $C_{Qq}^{3,1}/\Lambda^2$ set to different values. From these samples, the expected relative change in the signal event yield is derived as a function of $C_{Qq}^{3,1}/\Lambda^2$. The obtained parameterisation is used to perform a binned maximum likelihood fit. Finally, a likelihood scan is performed to extract the 95% confidence interval on $C_{Qq}^{3,1}/\Lambda^2$. The obtained observed confidence interval on $C_{Qq}^{3,1}/\Lambda^2$ is

$$-0.37 \leq C_{Qq}^{3,1}/\Lambda^2 \leq 0.06, \tag{1}$$

and therefore compatible with the Standard Model.

## 3.2 CKM interpretation

Both the production and decay of single top quarks involve a $Wtq$ vertex, whose cross-section is proportional to $f_{LV}^2 \cdot |V_{tq}|^2$. $V_{tq}$ denotes the respective CKM matrix elements with $q \in (b,d,s)$ and $f_{LV}$ is the left-handed form factor, which is exactly one in the Standard Model. The total number of expected signal events is the sum over all combinations of possible production and

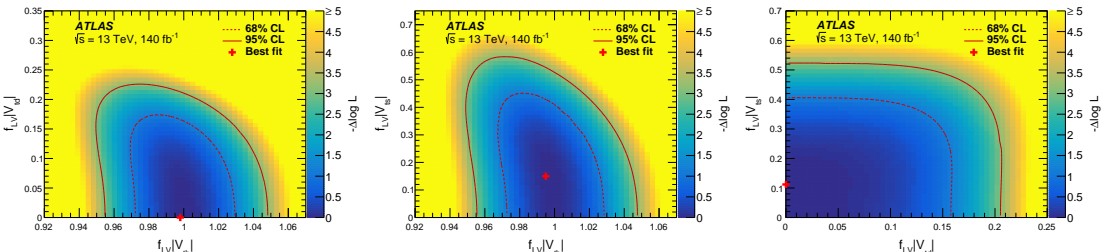

Figure 3: Confidence contours obtained from the two-dimensional likelihood scans. The difference of the log-likelihood function for points in the two-dimensional plane is taken with regard to the minimum of the function, indicating the best fit result [2].

decay channels

$$N_{\text{sig}} = \sum_{i=1}^{3} \sum_{j=1}^{3} N_{\text{sig},ij}, \text{ with}$$

$$N_{\text{sig},ij} = \mathcal{L} \cdot \underbrace{\sigma_t^i |V_{ti}|^2}_{\text{prod.}} \cdot \underbrace{\mathcal{B}(t \to jW)}_{\text{decay}}, \qquad (2)$$

with $\sigma_t^i = \sigma_t(V_{ti} = 1)$, flavour indices $i, j = b, d, s$.

This relation is used to constrain the individual matrix elements $V_{tq}$. Dedicated MC samples are produced for the single-top $t$-channel and $t\bar{t}$ processes, modeling the events for all combinations of $Wtq$ vertices in the top-quark production and decay. The limits on the individual matrix elements are set via two-dimensional likelihood scans. For each scan, one parameter is fixed to a certain value, which is zero for $f_{LV} \cdot |V_{td}|$, $f_{LV} \cdot |V_{ts}|$ and one for $f_{LV} \cdot |V_{tb}|$. The results of the scans are presented in Figure 3.

## 4 Conclusion

The production cross-section of single top-quarks produced via the $t$-channel process is measured at the LHC by ATLAS in proton-proton collisions at a center-of-mass energy $\sqrt{s} = 13$ TeV. The cross sections for single top-quark and single top-antiquark production are measured to be $\sigma_{tq} = 137^{+8}_{-8}$ pb and $\sigma_{\bar{t}q} = 84^{+6}_{-5}$ pb. For the combined cross section and the ratio $R_t$, $\sigma_{tq+\bar{t}q} = 221^{+13}_{-13}$ pb and $R_t = 1.636^{+0.036}_{-0.034}$ are measured. This measurement provides the most precise results of the process under study to this date. The theory predictions calculated at NNLO are in good agreement with the results.

As interpretations of the measurement, the impact of the EFT operator $O_{Qq}^{3,1}$ is constrained by setting limits on the Wilson coefficient $C_{Qq}^{3,1}/\Lambda^2$. A 95% confidence interval $-0.37 \leq C_{Qq}^{3,1}/\Lambda^2 \leq 0.06$ is obtained. The parameters $f_{LV} \cdot |V_{td}|$, $f_{LV} \cdot |V_{ts}|$ and $f_{LV} \cdot |V_{tb}|$ are constrained via two-dimensional likelihood-scans. All interpretations yield results compatible with the Standard Model.

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
