# Peer review of "Measurement of the $t$-channel single top-quark production cross section at $\sqrt{s}=13$ TeV with the ATLAS detector and interpretations of the measurement"

_SciPost Physics Proceedings_

## Round 1 · Referee Report · Anonymous (Referee 1) · 2024-12-5

Strengths

The results provide the most precise t-channel single top-quark production cross sections at 13 TeV

Weaknesses

The length of these proceedings do not allow giving details of the analyese performed added as references.

Report

These proceedings described the ATLAS measurement of the t-channel single top-quark at a center-of-mass energy of 13 TeV presented at the TOP2024 conference.

Requested changes

  • Place figures after the reference, close to the text where the figures are described.
  • Caption figure 1, define Dnn
  • Section 2, "A good agreement between the MC simulation and the data is reached." Are all contributions estimated by MC? Figure 2 mentions predictions, if some contributions are data-driven, the sentence needs to be corrected accordingly.
  • Section 3.1, mention to what is the binned maximum likelihood fit performed to, is it also in the SRs and Dnn?
  • Figure 3, if space permits increase the size of the figures, it is difficult to see the best fit marks.
  • Section 4, "The theory predictions calculated at NNLO are in good agreement with the results.", add "except for ABMP and NNPDF3.1"

Recommendation

Ask for minor revision

---

## Editorial Decision

accepted_in_target_journal